# The Composite TiO_2_–CuO_x_ Layers Formed by Electrophoretic Method for CO_2_ Gas Photoreduction

**DOI:** 10.3390/nano13142030

**Published:** 2023-07-08

**Authors:** Larisa I. Sorokina, Andrey M. Tarasov, Anastasiya I. Pepelyaeva, Petr I. Lazarenko, Alexey Yu. Trifonov, Timofey P. Savchuk, Artem V. Kuzmin, Aleksey V. Tregubov, Elena N. Shabaeva, Ekaterina S. Zhurina, Lidiya S. Volkova, Sergey V. Dubkov, Dmitry V. Kozlov, Dmitry Gromov

**Affiliations:** 1Institute of Advanced Materials and Technologies, National Research University of Electronic Technology—MIET, Bld. 1, Shokin Square, Zelenograd, 124498 Moscow, Russia; 2Scientific Research Institute of Physical Problems Named after F.V. Lukin, Pass. 4806, Bld., Zelenograd, 124498 Moscow, Russia; 3S.P. Kapitsa Scientific Technological Research Institute, Ulyanovsk State University, 42 Leo Tolstoy Street, 432017 Ulyanovsk, Russia; 4Institute of Nanotechnology of Microelectronics RAS, 32A Leninsky Prospekt, 119991 Moscow, Russia; 5Institute for Bionic Technologies and Engineering, I.M. Sechenov First Moscow State Medical University, Bolshaya Pirogovskaya 2, 119435 Moscow, Russia

**Keywords:** electrophoretic deposition, TiO_2_–CuO_x_ composites, photocatalysis, CO_2_ photoreduction

## Abstract

This study demonstrates the ability to control the properties of TiO_2_–CuO_x_ composite layers for photocatalytic applications by using a simple electrophoretic deposition method from isopropanol-based suspension. To obtain uniform layers with a controlled composition, the surfactant sodium lauryl sulfate was used, which influenced the electrophoretic mobility of the particles and the morphology of the deposited layers. The TiO_2_–CuO_x_ composite layers with different CuO_x_ contents (1.5, 5.5, and 11 wt.%) were obtained. It is shown that the optical band gap measured by UV–VIS–NIR diffuse reflectance spectra. When CuO_x_ is added to TiO_2_, two absorption edges corresponding to TiO_2_ and CuO_x_ are observed, indicating a broadening of the photosensitivity range of the material relative to pure TiO_2_. An open-circuit potential study shows that by changing the amount of CuOx in the composite material, one can control the ratio of free charge carriers (n and p) and, therefore, the catalytic properties of the material. As a result, the TiO_2_–CuO_x_ composite layers have enhanced photocatalytic activity compared to the pure TiO_2_ layer: methanol yield grows with increasing CuO_x_ content during CO_2_ photoreduction.

## 1. Introduction

The search for strategies to promote the formation of heterostructured photocatalysts is an urgent task to improve the environmental situation in the world through the production of renewable fuels (such as hydrogen, methanol, and methane) and the decomposition of toxic pollutants [1,2,3]. The development of hybrid photocatalytic systems based on TiO_2_ in combination with other semiconductors with a narrower band gap is one of the most economical and reasonable ways to increase the efficiency [4,5,6,7]. Improved photocatalytic characteristics of TiO_2_- and CuO_x_-based composites, which demonstrate activity in the visible radiation range, are presented in [8,9,10]. The TiO_2_/CuO_x_ heterostructure promotes a more efficient photon collection and separation of electrons and holes through the interfacial charge transfer process [11]. To date, there is a wide variety of methods for the formation of photocatalytic layers on a solid surface, including chemical vapor deposition (CVD), pulsed laser deposition (PLD), spray pyrolysis, electrochemical deposition, anodic oxidation, hydrothermal method, magnetron sputtering, sol–gel, electrophoretic deposition (EPD), and others.

Electrophoretic deposition technology is an appealing manufacturing method. Its principle of operation is based on the movement of charged particles in a suspension under the influence of an electric field. This EPD technology does not require expensive equipment and allows for the formation of composites with complex compositions on large-area substrates while accurately controlling geometric and stoichiometric parameters. It is also worth noting that the EPD technology allows for the use of commercially available powders, such as TiO_2_ P25, whose effectiveness has been demonstrated in numerous papers [12,13,14,15]. The papers [16,17] present specific features of TiO_2_ P25 layer formation by the EPD method.

The formation of composite layers by electrophoretic deposition is noticeably more complicated since particles of different compositions differ in zeta potential in the same solvent. Heterogeneous particles can migrate to the electrode individually with different electrophoretic mobility or coagulate with each other, forming heteroaggregates [18]. To stabilize the multi-component suspension, various charging, binding, and dispersing agents are used, which prevent the formation of large aggregates and provide a high surface charge of particles. Many studies focus on the selection of optimal additives for electrophoretic deposition of composites with TiO_2_ [19,20,21].

In this work, the surfactant sodium lauryl sulfate (SDS) was used as a stabilizing additive for the deposition of TiO_2_ (P25) and TiO_2_–CuO_x_ composite layers. The main goal of the study was to develop a simple and efficient technique for forming composite layers based on TiO_2_ and CuO_x_ with controlled composition for potential application as photocatalysts. Optical band gap, open-circuit potential, and photocatalytic activity of the developed composite photocatalysts were investigated. The enhanced photocatalytic CO_2_ conversion to methanol by TiO_2_–CuO_x_ composites, as compared to pure TiO_2_, is explained by the broadening of the spectral range of photoactivity, improved charge carrier separation, and increased redox capacity.

## 2. Materials and Methods

### 2.1. Materials and Electrophoretic Deposition Process

CuO nanopowder with an average particle size of 50–100 nm containing a small amount of Cu_2_O phase (Advanced Powder Technologies LLC, Tomsk, Russia), TiO_2_ P25 nanopowder (Evonik, Hanau, Germany) with an average particle size of 25 nm, sodium lauryl sulfate, and chemically pure isopropyl alcohol were used for the suspension preparation. The loading of TiO_2_ and TiO_2_/CuO_x_ nanopowders in all cases was 1 g/L. The 50 mL suspension was processed using a 100 watt ultrasonic homogenizer for 1 h at a temperature not exceeding 22 °C. Titanium foil was used as a substrate. Stainless steel foil was used as a counter-electrode. Preliminarily titanium foil was etched in HF:HNO_3_:H_2_O solution (1:1:4 by volume) to remove the impurities and titanium oxide layer. Then the foil was washed in deionized water and dried in vapor of isopropyl alcohol. The EPD process was carried out in a potentiometric mode at an electric field strength from 30 to 130 V/cm.

### 2.2. Material Characterization

Surface morphology and composition of received TiO_2_–CuO_x_ layers were investigated by scanning electron microscopy Helios G4 CX (FEI, Hillsboro, Oregon, USA) and energy-dispersive X-ray (EDX) analysis. The study of suspension particle aggregates was carried out using Tecnai G^2^ 20 electron microscope (FEI, Hillsboro, Oregon, USA), at an accelerating voltage of 200 kV. The microscope is equipped with a system for energy-dispersive X-ray analysis.

The phase composition of the obtained TiO_2_–CuO_x_ layers was studied using X-ray diffraction. Diffraction wide-angle spectra were recorded by Miniflex 600 (Rigaku, Tokyo, Japan) under the following conditions:Scanning speed 10 °C·min^−1^;Scan range from 3° to 90°;Radiation Cu K_α;_The voltage on the tube 20 Kv;The anode current 7 mA.

### 2.3. Optical Band Gap Investigation

The optical band gap (E_g_) was estimated from UV–VIS–NIR diffuse reflectance spectra. The diffuse reflectance spectra of the powder samples of TiO_2_–CuO_x_ were measured using a spectrophotometer Agilent Cary 5000 (Santa Clara, California, USA) with an integrating sphere. Reflectance spectra were recorded in the range from 200 to 800 nm with a spectral resolution of 1 nm. The diffuse reflectance data have been transformed into an absorbance according to the Kubelka–Munk Equation (1) [22,23,24,25,26]:(1)$$F\left(R_{d}\right)=\frac{{\left(1-R_{d}\right)}^{2}}{2R_{d}}$$
where R_d_ is the diffuse reflectance and F(R_d_) is the function, which is proportional to the absorbance.

To estimate the optical band gap (E_g_^opt^), we used Equation (2) [26,27,28]:(2)$$F\left(R_{d}\right)\cdot h\cdot \nu =A\cdot {\left(h\cdot \nu -{E_{g}}^{opt}\right)}^{n}$$
where n is a factor characterizing the type of transition in the forbidden zone. We used n = 2, which is typical for the indirect allowed transition [24]. The optical band gap values were obtained from the linear fits to (F(R_d_) · hν)^1/2^ versus hν (frequently called Tauc plot) [29,30,31,32].

### 2.4. Photoelectrochemical Measurements

The dependence of open-circuit potential (OCP) on time was recorded using an Electrochemical Instruments P-45X potentiostat (Chernogolovka, Russia). The open-circuit potential was measured relative to the Ag/AgCl (3 M) reference electrode in a 0.1 M aqueous Na_2_SO_4_ solution. A platinum sheet with a total area of 2 cm^2^ was used as a counter-electrode. Measurements were performed under the full-spectrum illumination of a xenon lamp. The Newport 67005 Sun complex with Xe 150 W lamp (Irvine, CA, USA) was used as a light source. The power of the incident light on the sample was approximately 100 mW/cm^2^, and the illuminated area was approximately 0.63 cm^2^. We waited until the OCP_dark_ value reached a stable state before illuminating the samples. The samples were illuminated for 200 s.

### 2.5. Photocatalytic Activity Study

For photocatalytic studies, TiO_2_–CuO_x_ layers with 7 cm^2^ area and specific mass of 1 mg/cm^2^ were deposited on a titanium foil substrate. The photocatalytic studies of CO_2_ reduction were conducted in a 25 cm^3^ flow reactor equipped with a heating element and a quartz window. Two 35 W xenon lamps were used as the light source. The process conditions were as follows: the reactor temperature was 40 °C, the reaction mixture consisted of 40 vol.% CO_2_ and 60 vol.% H_2_O, and the gas flow rate was 3.0 mL/min. Prior to injecting CO_2_ into the reaction chamber, the gas line was purged with helium. Subsequently, the sample was illuminated for 12 h in a gas flow consisting of 40 vol.% He and 60 vol.% H_2_O to decompose the adsorbed organics. Analysis of reaction products was performed on the gas chromatograph Crystal 5000 (Chromatec, Yoshkar-Ola, Russia) equipped with an Agilent HP PLOT Q capillary column and a flame ionization detector. Identification of reaction products and measurement of their concentration were carried out based on the exit time and peak area, respectively. Control measurements without light irradiation were performed after changing the He gas flow to CO_2_. For all samples, three control measurements of the methanol yield without light were performed, and the average value was calculated, which was considered as the baseline value. All presented methanol yield values were obtained by subtracting the baseline methanol yield value in the dark. The methanol yield concentration (mg/m^3^) was calculated from the calibration curve. To construct the curve, mixtures with different concentrations of methanol were prepared and the peak area of methanol (mV·min) was calculated as a function of concentration. The final methanol yield (μmol/m^2^·h) of the sample was calculated according to Equation (3):(3)$$Yield(\mu mol/m^{2}\cdot h)=\frac{C(mg/m^{3})\cdot Q(m^{3}/h)}{M(mg/\mu mol)\cdot A(m^{2})}$$
where C—concentration, Q—flow rate, M—molar mass, and A—sample area.

## 3. Results

### 3.1. Electrophoretic Deposition of TiO_2_ Layers

The suspension based on isopropanol and TiO_2_ particles demonstrates high stability: the particles remain suspended for one day. However, electrophoretic deposition from this suspension practically does not occur. This indicates the low surface charge of the particles, despite the fact that the particles are sterically stabilized. To optimize the EPD process, the effect of the surfactant sodium lauryl sulfate on the electrophoretic deposition rate of TiO_2_ particles and on the morphology of the deposited layers was studied. The correlation between the particle electrophoretic deposition rate and the SDS content in the suspension is shown in Figure 1.

With an increase in SDS content in the suspension, the electrophoretic deposition rate of TiO_2_ particles reaches a certain maximum value and then decreases. The maximum deposition rate is observed at contents in the range of 0.1–0.14 mg of SDS per 1 mg of TiO_2_ in an isopropyl alcohol-based suspension. The maximum deposition rate is observed at a content of 0.1 and 0.14 mg SDS per 1 mg TiO_2_ in a suspension based on isopropyl alcohol. With an SDS content of 0.2 mg per 1 mg of TiO_2_, a critical concentration of surfactants in a colloidal solution is achieved. As a result, the particles coagulate and the deposition rate decreases, since it is more difficult for larger aggregates to migrate to the electrode. The comparison of the deposited layers at 90 V/cm for 5 min from suspensions with different SDS contents is shown in Figure 2.

Uniform TiO_2_ layers were obtained from suspensions that demonstrated the highest rate of electrophoretic particle deposition. When the SDS content is less than 0.1 mg per 1 mg of TiO_2_, deposition occurs with agglomerates (Figure 2a). At SDS content greater than 0.14 mg per 1 mg of TiO_2_, the layer forms unevenly and with low adhesion (Figure 2d). Taking into account the deposition rate and uniformity of the deposited layers, favorable ratios are within the range of 0.1 mg to 0.14 mg of SDS per 1 mg of TiO_2_. In this case, we consider the ratio of 0.1 mg SDS per 1 mg TiO_2_ for the formation of titanium-dioxide-based layers to be more preferable. This choice is based on the belief that a lower content of SDS will introduce fewer impurities into the formed layers.

### 3.2. Electrophoretic Deposition of TiO_2_–CuO_x_ Layers

No additional stabilizing additives are required for electrophoretic deposition of CuO_x_ from an isopropanol-based suspension. When CuO_x_ and TiO_2_ particles are mixed, the suspension remains stable for one day, but only CuO_x_ particles are electrophoretically deposited. A similar study of the influence of SDS on the electrophoretic deposition rate and the morphology of the deposited layers was conducted. When SDS is added to the suspension containing CuO_x_ and TiO_2_ particles, the rate of electrophoretic deposition also increases up to a certain maximum value and then decreases. However, in this case, a significantly lower amount of SDS is required (0.04 mg SDS per 1 mg of TiO_2_ nanopowder), because CuO_x_ particles do not require additional electrostatic stabilization.

A series of samples deposited from suspensions containing 3%, 10%, and 20% wt. CuO_x_ were obtained. Figure 3 shows SEM images and EDX element mapping of the TiO_2_–CuO_x_ layer deposited from a suspension with 20 wt.% CuO_x_. The composition of the resulting TiO_2_–CuO_x_ composite layers was determined by EDX analysis. (Table 1).

The TiO_2_–CuO_x_ layer formed by electrophoretic deposition has a developed surface (Figure 3a). All elements are distributed uniformly over the entire area, which is confirmed by Figure 3b,d. The approximate thickness of the TiO_2_–CuOx layer with a specific mass of 1 mg/cm^2^ is 8 μm (Figure 3c).

The mass content of CuO_x_ in the layer differs from the content of CuO_x_ in the suspension by almost two times (Table 1). Aggregates of particles from suspensions with the addition of SDS were studied using a transmission electron microscope (Figure 4).

To identify TiO_2_ and CuO_x_ particles, a characteristic fragment of the investigated sample was selected, shown in Figure 4a,b. In STEM mode, the high-angle annual dark-field detector (HAADF) was used. Therefore, the light parts of the image in Figure 4b correspond to the fragments of the sample that cause maximum scattering of the electron beam. The images show two large particles surrounded by small ones. EDX maps of this fragment (Figure 4c,d) show that large particles correspond to copper oxide, and small particles correspond to titanium oxide. Figure 4f shows that smaller TiO_2_ particles (~25 nm) are concentrated around large CuO_x_ particles (~50–100 nm), forming TiO_2_–CuO_x_ heteroaggregates. It is assumed that TiO_2_–CuO_x_ heteroaggregates and individual TiO_2_ particles migrate during the EPD process. The size of the TiO_2_–CuO_x_ heteroaggregates is ten times larger than the size of TiO_2_ particles. This facilitates slower electrophoretic deposition of copper oxide particles.

The phase composition of the deposited TiO_2_–CuO_x_ layers was investigated by XRD (Figure 5).

Figure 5 shows that the XRD pattern indicates the presence of three phases: anatase, rutile, and CuO. Phases of anatase and rutile correspond to commercial TiO_2_ P25 nanopowder. We could not identify the Cu_2_O phase, which is present in small amounts in the commercial powder CuO_x_ in accordance with the certificate. Thus, the electrophoretic deposition process of TiO_2_–CuO_x_ composites as a whole does not change the phase composition relative to the initial powders.

Figure 6 shows the dependences of the EPD rate on the electric field strength for TiO_2_–CuO_x_ layers with CuO_x_ content of 1.5; 5.5; and 11 wt.% (deposition time 2 min).

The deposition rate of TiO_2_–CuO_x_ layers increases linearly with increasing electric field strength. With an increase in the content of CuO_x_ particles, the deposition rate of the TiO_2_–CuO_x_ layer decreases, which also confirms that the electrophoretic mobility of TiO_2_–CuO_x_ heteroaggregates is lower. The optimal range of electric field voltage for TiO_2_–CuO_x_ formation is determined to be above 110 V/cm, as it enables the deposition of layers with a high deposition rate. For subsequent studies, TiO_2_–CuO_x_ layers with a specific mass of 1 mg/cm^2^ were deposited at an electric field strength of 110 V/cm, while varying the duration of the EPD process.

### 3.3. Optical Band Gap

Knowledge of the optical bandgap (E_g_^opt^), which characterizes the optoelectronic properties and determines the position of the absorption edge, is crucial for purposeful optimization of the photocatalytic properties of the formed composite materials.

The estimation of the optical band gap of the formed composite TiO_2_–CuO_x_, pure TiO_2_ P25, and pure CuO_x,_ determined by linear extrapolation of the absorption edges on the dependence of (F(R_d_)hν)^1/2^ on hν, is presented in Figure 7 and Table 2.

Determination of the optical bandgap for pure TiO_2_ and CuO_x_ by absorption edge is not problematic and gives a value of approximately 3.08 eV and 1.36 eV, respectively. In the case of adding CuO_x_ to TiO_2_, the dependences (F(R_d_)hν)^1/2^ on E show two absorption edges, obviously corresponding to TiO_2_ and CuO_x_. Herewith, applying the approach of determining the optical band gap of TiO_2_ by the intersection of the tangent with the abscissa axis shows a monotonic decrease in the determined value of E_g_^opt^ (see E_g21_^opt^ in Table 2). A similar decrease effect has been observed in other works [33,34]. This is probably due to the combination of the absorption spectra of two materials: TiO_2_ and CuO_x_. A solution to this problem was described in [35,36], where the diffusion spectra of samples consisting of a combination of two materials were analyzed. In these works, the estimation of E_g_^opt^ is performed by plotting two tangents to the fundamental absorption edge and the slope below the fundamental absorption. The intersection point of the linear fits can be interpreted as optical band gap value. Using this approach in our case gives an increase in the E_g_^opt^ values for TiO_2_ (see E_g22_^opt^ in Table 2) with increasing CuO_x_ content, as can be seen in Figure 7 also. Despite this, the application of the second approach gives values E_g_^opt^ closer to pure TiO_2_.

Thus, the two observed absorption edges indicate that the addition of CuO_x_ to TiO_2_ enables the extension of the photosensitivity range of the material.

### 3.4. Open-Circuit Potential

The TiO_2_–CuO_x_ composite layers were further analyzed using chronopotentiometry in a photoelectrochemical cell. Figure 8 presents the dependencies of the open-circuit potential (OCP) relative to the Ag/AgCl reference electrode on the illumination time of the TiO_2_–CuO_x_ composites.

The open-circuit potential is the potential difference between the working electrode and the reference electrode when there is no current flowing through the external circuit. When the photoelectrode is illuminated, any change in the OCP reflects the generated photovoltage [37]. The value of the photovoltage of a semiconductor electrode represents the difference of the Fermi quasi-levels of holes (Ef,p) and electrons (Ef,n) and is directly related to the concentration of free charge carriers in the semiconductor [38]. The photogenerated charge carriers accumulate in the photoelectrode material and the change in OCP depends on the ratio of n and p [39]. The negative photovoltage indicates the anodic properties of the photoelectrode, while the positive photovoltage indicates the cathodic properties characteristic of the n-type and p-type materials [37,40].

Figure 8b shows that the OCP light/dark change of the sample with pure TiO_2_ is negative (−57 mV), which indicates that there are more electrons than holes (n ≫ p) and that the electrode based on this material is the anode. However, for samples with 1.5, 5.5, and 11 wt.% CuO_x_ content, the change in OCP becomes more positive and increases with CuO_x_ content in electrode material (−50, 1 and 3.2 mV, respectively). As mentioned above, the sample of pure TiO_2_ exhibits anodic properties, but with increasing content of copper oxide in the photoelectrode material, the samples begin to exhibit more cathodic properties, which can be attributed to a change in the ratio of free charge carriers (n and p).

Thus, by changing the amount of CuOx in the composite material, it is possible to control the ratio of photogenerated n and p carriers, and, consequently, the catalytic properties of the material.

### 3.5. Photocatalytic Activity

We investigated the photocatalytic activity of TiO_2_–CuO_x_ composite layers deposited on titanium foil. The main product of CO_2_ conversion under full-spectrum irradiation is methanol. Figure 9 shows the dynamics of methanol yield over time (a) and the average methanol yield as a function of the copper oxide content in the layer (b).

It can be seen from Figure 9 that the methanol yield increases almost linearly with increasing content of copper oxide in the composite material. The highest methanol yield is approximately 3.4 μmol/m^2^·h for the sample with 11 wt.% CuO_x_, which is two times higher compared to pure TiO_2_ P25. As we know [41] and observe, pure TiO_2_ P25 can produce methanol through the generation of electron–hole pairs under the influence of light (UV) in the probable scheme:(4)$$H_{2}O+h^{+}\to H^{+}+{OH}^{\cdot }$$
(5)$${CO}_{2}+6H^{+}+6e^{-}\to {CH}_{3}OH+H_{2}O$$

The addition of p-type CuO_x_ to n-type TiO_2_ and the formation of a composite material lead to the local appearance of both a heterojunction and a p–n junction at the same time. This provides at least two advantages: (i) more efficient use of light-generated charge carriers due to their increased lifetime, which is achieved through the separation of charge carriers by the p–n heterojunction; (ii) expansion of the spectral range of photoactivity of the material, since TiO_2_ P25 is photoactive in the UV region, while copper oxide, having a narrower band gap, is photoactive in the visible spectrum.

A probable scheme of charge carrier separation during irradiation of the TiO_2_/CuO_x_ material is presented in the papers [42,43,44,45,46]. We assume that this mechanism is realized in our case and is shown in Figure 10.

Under the influence of radiation, electron–hole pairs are generated in both materials. A probable process involves the recombination of electrons from the conduction band (CB) of TiO_2_ with the holes of CuO in the valence band (VB) of CuO (Figure 10). As a consequence, holes accumulate in VB of TiO_2_ and electrons accumulate in the CB of CuO. This leads to the oxidation of adsorbed H_2_O molecules on the TiO_2_ surface, producing the required H+ ions for reaction (5), and the photoreduction of adsorbed CO_2_ occurs on CuO. Therefore, increasing the amount of copper oxide in the composite results in an increase in the number of catalytic centers with CO_2_ adsorption, thereby intensifying the reaction (5).

## 4. Conclusions

A technique for the formation of TiO_2_ and composite TiO_2_–CuO_x_ layers by electrophoretic deposition from a suspension based on isopropanol with the addition of SDS was developed. Composite layers based on TiO_2_ P25 and CuO_x_ in a certain ratio can be easily obtained by the EPD method. By varying the ratio of TiO_2_ and CuO_x_ it is possible to adjust the optical and photocatalytic properties. The addition of CuO_x_ to TiO_2_ results in two absorption edges corresponding to TiO_2_ and CuO_x_ to indicate a photosensitivity range expansion of composite TiO_2_–CuO_x_ material in comparison to pure TiO_2_. Altering the quantity of CuO_x_ in the composite material allows for the manipulation of the proportion of free charge carriers (n and p), as evidenced by an open-circuit potential investigation. Thus, the catalytic properties of the material are controlled. The photocatalytic activity of the formed composite layers was investigated using the reaction of CO_2_ reduction to methanol as a model reaction. It is found that the methanol yield increase linearly with increasing content of copper oxide in the composite material. The highest methanol yield is approximately 3.4 μmol/m^2^·h for the sample with 11 wt.% CuO_x_, which is two times higher compared to pure TiO_2_ P25. The increased photocatalytic activity of TiO_2_–CuO_x_ composite layers in comparison with pure TiO_2_ is explained by the formation of a p–n heterojunction, which expands the spectral range of photoactivity, separates photogenerated charge carriers, and enhances the redox capacity.

## Figures and Tables

**Figure 1 nanomaterials-13-02030-f001:**
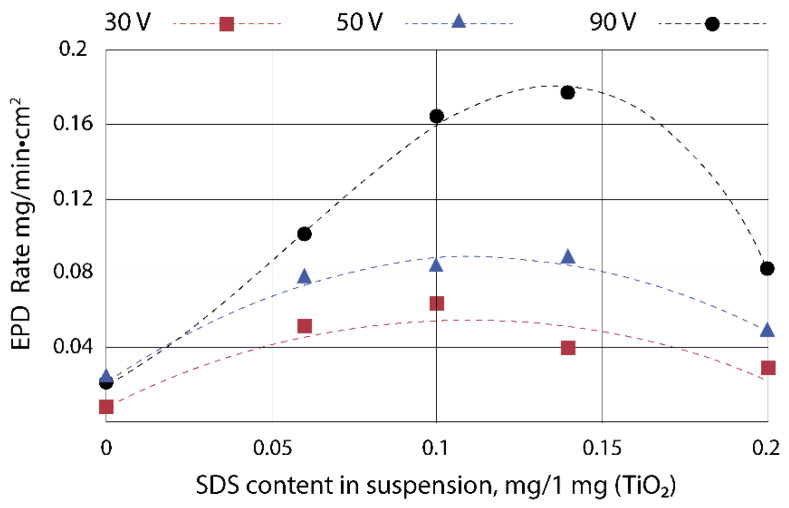
EPD rate of TiO_2_ particles as a function of SDS content per 1 mg of TiO_2_ in suspension at different electric field strengths.

**Figure 2 nanomaterials-13-02030-f002:**
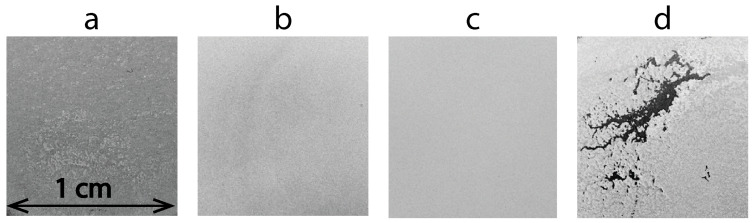
Photo images of TiO_2_ layers deposited from suspension with different contents of SDS per 1 mg of TiO_2_: (**a**) 0.06 mg, (**b**) 0.1 mg, (**c**) 0.14 mg, (**d**) 0.2 mg.

**Figure 3 nanomaterials-13-02030-f003:**
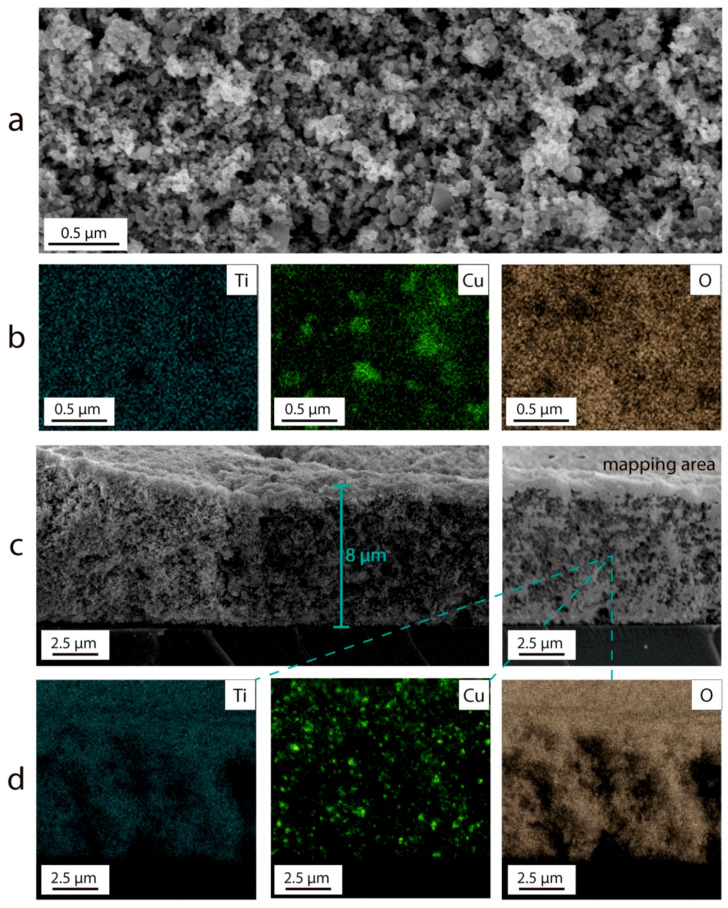
SEM image of layer surface (**a**), element mapping of layer surface (**b**) cross-section (**c**), cross-section element mapping (**d**) of TiO_2_–CuO_x_ deposited from suspension with 20 wt.% CuO_x_.

**Figure 4 nanomaterials-13-02030-f004:**
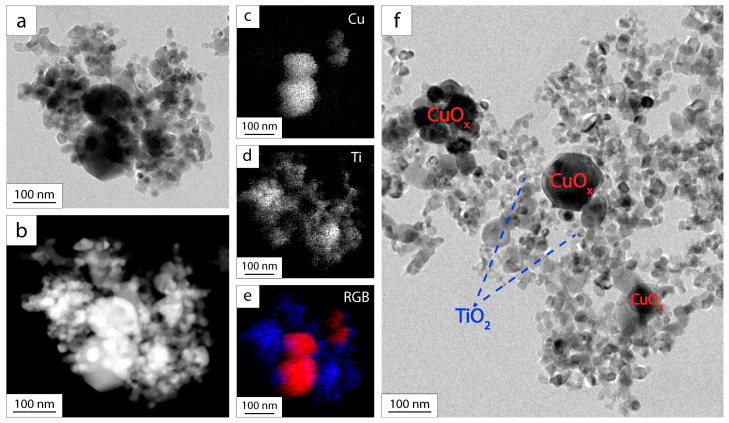
TEM images of TiO_2_–CuO_x_ aggregates in suspension with the addition of SDS: TEM (**a**) and STEM HAADF (**b**) images of a fragment of the aggregate; EDX maps of copper (**c**) and titanium (**d**) distribution; combined RGB image (**e**): Cu—red; Ti—blue; TEM image of the aggregate (**f**).

**Figure 5 nanomaterials-13-02030-f005:**
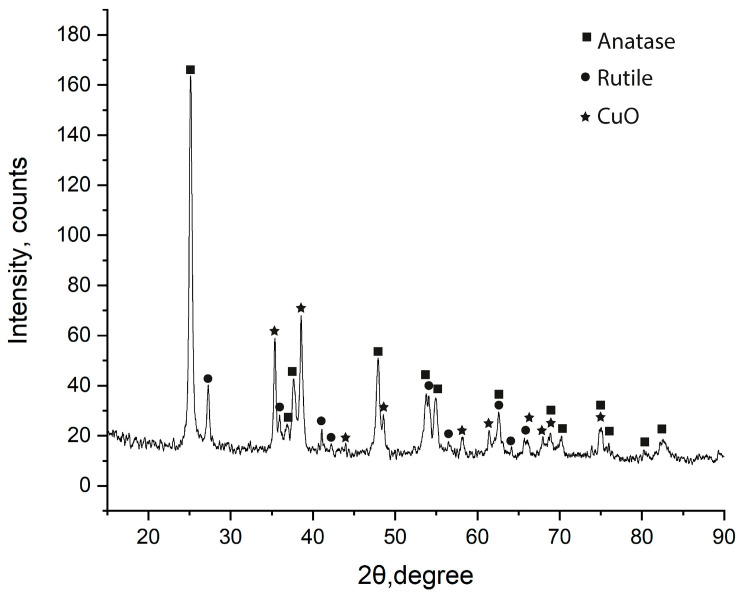
XRD pattern of TiO_2_–CuO_x_ layers with 11 wt.% CuO_x_ content.

**Figure 6 nanomaterials-13-02030-f006:**
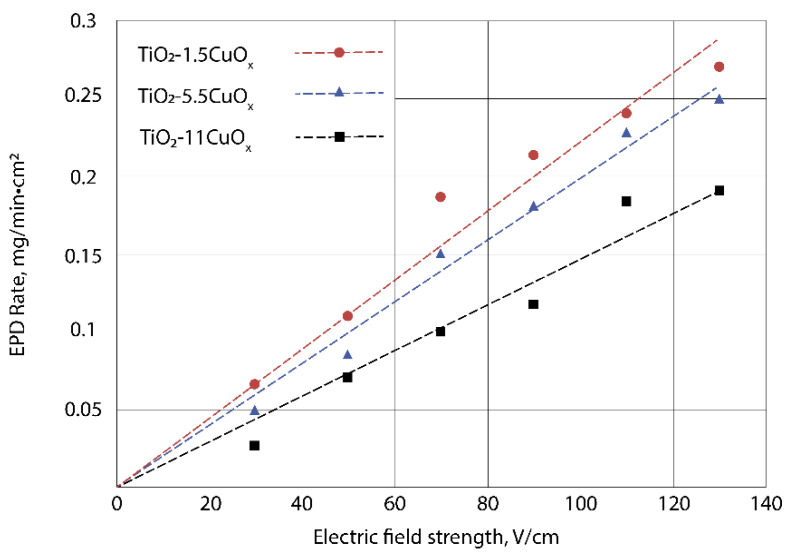
Dependences of EPD rate of TiO_2_–CuO_x_ layers on electric field strength.

**Figure 7 nanomaterials-13-02030-f007:**
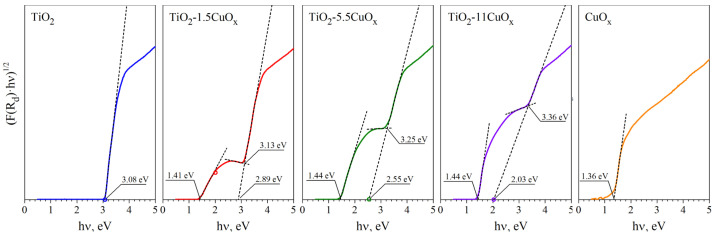
Determination of the optical band gap of composites with different CuO_x_ contents, colored solid line—experiment, dash line—linear fit.

**Figure 8 nanomaterials-13-02030-f008:**
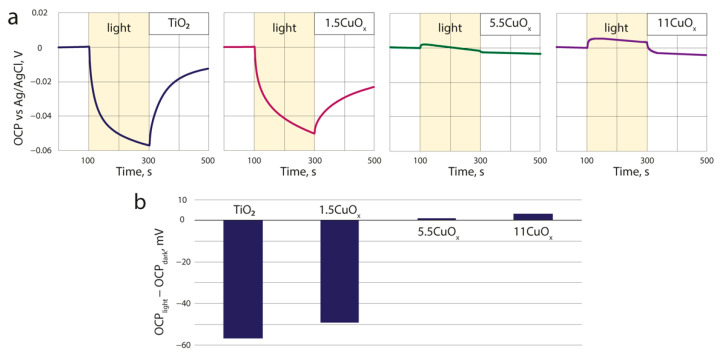
Dependences of OCP on illumination time (**a**), OCP_light_ − OCP_dark_ difference diagram of TiO_2_–CuO_x_ composites (**b**).

**Figure 9 nanomaterials-13-02030-f009:**
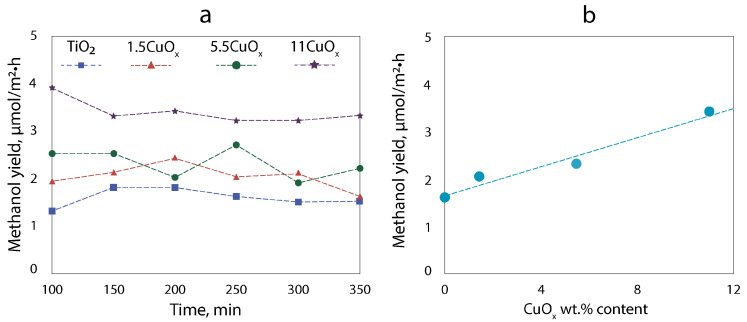
Methanol yield over time (**a**) and the average methanol yield as a function of the copper oxide content in the layer (**b**).

**Figure 10 nanomaterials-13-02030-f010:**
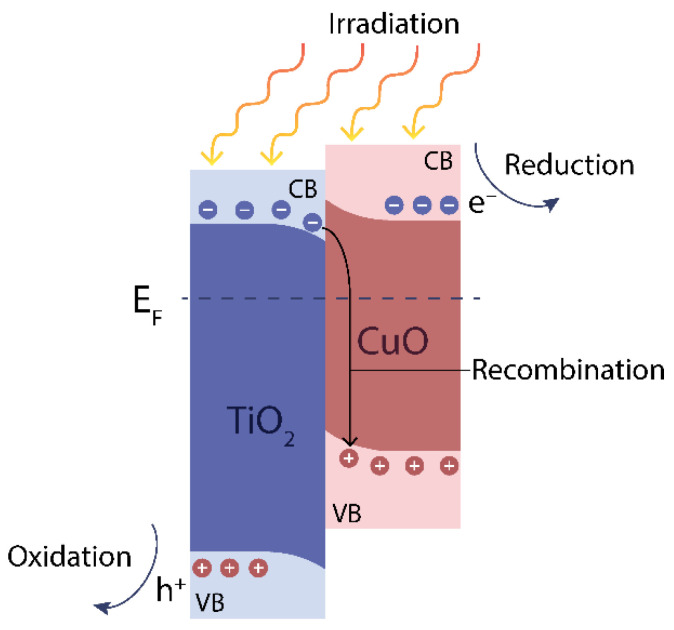
Schematic illustration of the e^−^/h^+^ pair generation mechanism at the TiO_2_–CuO contact.

**Table 1 nanomaterials-13-02030-t001:** CuO_x_ content, wt.% in the deposited layer as a function of CuO_x_ wt.% in the suspension.

CuO_x_ wt.% in the Suspension	CuO_x_ wt.% in the Deposited Layer
3	1.5
10	5.5
20	11

**Table 2 nanomaterials-13-02030-t002:** Optical band gap of TiO_2_ and TiO_2_–CuO_x_ layers formed by EPD method.

CuO_x_ Content, wt.%	E_g1_^opt^, eV	E_g21_^opt^, eV	E_g22_^opt^, eV
0	-	3.08	3.08
1.5	1.41	2.89	3.13
5.5	1.44	2.55	3.25
11	1.44	2.03	3.36
100	1.36	-	-

## Data Availability

The data presented in this study are available upon request from the corresponding author.

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
