# Peer review of "The Composite TiO2–CuOx Layers Formed by Electrophoretic Method for CO2 Gas Photoreduction"

_nanomaterials, 2023, doi:10.3390/nano13142030_

Round 1

Reviewer 1 Report

nanomaterials-2472735

Title: The composite TiO2-CuOx layers formed by electrophoretic method for CO2 gas photoreduction

The manuscript describes the synthesis and characterization of TiO2-CuOx composites, with different CuOx content, obtained by electrophoretic method. The photocatalytic activity in the photoreduction of CO2 at methanol was determined.

The manuscript is generally well-written, clear and concise.  I have the following observations:

1.     The authors must describe how was determined the quantity of methanol. I can not find any information about the methanol determination. This is a major observation, because the determination of photocatalytic activity is based on the quantitative information for the methanol synthesis.

2.     A reference for the Kubelka-Munk method is necessary. Why was this method actually mentioned if the Tauc model is used?

3.     “Tautz method” must be “Tauc method”. The error is repeated throughout the manuscript.

4.     The following information is repeated, with a difference. L 181-184: “The figure shows that smaller TiO2 particles (~25 nm) are concentrated around large CuOx particles (~50-100 nm), forming TiO2-CuOx heteroaggregates. It is assumed that TiO2-CuOx heteroaggregates and individual TiO2 particles migrate during the EPD process. The size of TiO2-CuOx heteroaggregates is ten times larger than the size of TiO2 particles. This facilitates slower electrophoretic deposition of copper oxide particles.”. L 186-189: “The figure shows that smaller TiO2 particles (~25 nm) are concentrated around large CuOx particles (~50-80 nm), forming TiO2-CuOx heteroaggregates. It is assumed that TiO2 CuOx heteroaggregates and individual TiO2 particles migrate during the EPD process. The size of TiO2-CuOx heteroaggregates is ten times larger than the size of TiO2 particles. This facilitates slower electrophoretic deposition of copper oxide particles.”. The difference is "CuOx particles (~50-100 nm)" vs. "CuOx particles (~50-80 nm)". It must be corrected.

Reviewer 2 Report

In the present study, the authors have synthesized TiO2-CuOx composites using electrophoretic method. The photocatalytic activity of composites have been tested by exploring  COgas photoreduction. After adjusting the proper growth parameters for the deposition of TiO2-CuOx, the composite materials have shown better efficiency in COgas photoreduction under UV-visible irradiation in comparison to pure TiO2.

The deposition method and experimental results are clearly presented and explained and the authors have shown that TiO2-CuOcomposites exhibit better photocatalytic activity than pure TiO2. Therefore, I find the paper to be relevant and I propose the publication of present study.

There are some minor issues. Firstly, some corrections in the text are necessary:

- page 4: the figure caption for Fig 2 is wrong (it is the same as in Fig. 1)

- page 6: the first paragraph on this page is repeated twice

My bigger issue is related to the chemical composition of the deposited material. The authors have used TiO2 and copper-oxide powders in their electrophoretic method, however they do not give the experimental results concerning the chemical state of the deposited material. Therefore, I strongly suggest that XPS or XRD measurements should be done on the TiO2-CuOcomposite material, confirming that there are no changea in the chemical composition of material during the chemical synthesis method.

Reviewer 3 Report

The manuscript entitled "The composite TiO2-CuOx layers formed by electrophoretic method for CO2 gas photoreduction". The TiO2-CuOx composites were fabricated by a simple electrophoretic deposition method from iso-propanol-based suspension has been performed. The surface morphology, optical properties and photocatalytic activity of as-prepared samples have been studied. Related comparative experiments were also performed. The experimental work is interesting. In my opinion, this work is interesting and has a certain reference in the development for the application in the fields of photocatalysis. However, there are some remarks that should be taken into consideration by the authors in order to raise this article to a good level for publication in Nanomaterials.

The suggested modifications are listed as follows:

1. “50 ml” should be revised as “50 mL.

2. “Tautz method” should be revised as “Tauc method”.

3. How did the author determine in Figure 4 that the large particle is CuOx and the small particle is TiO2? HRTEM should be performed to identify who is CuOx and who is TiO2.

4. The ordinate of Figure 6 should start from zero and the author should give it. If you don't start at 0, this calculation is wrong. At the same time, the optical bandgap value of CuOx should be given.

5. The photocurrent curve in Figure 7 looks very strange.

6. The conduction and valence band potentials of TiO2 and CuOx in Figure 9 should be calculated, and it is not enough to guess. Some literature can be referenced: https://doi.org/10.1016/j.apsusc.2022.154977.

Reviewer 4 Report

In this manuscript, the authors fabricated the TiO2-CuOx mixture layer by electrophoretic deposition method.  The photocatalytic activities of the TiO2-CuOx layer for CO2 photoreduction were evaluated.  The reviewer could not agree with many parts of the conclusions that the authors derived from the experimental results.  Therefore, the reviewer suggests rejection for the present study.  Detailed comments were listed below:

1.                 In Figure 3, please provide the cross-sectional SEM-EDS mapping of the photocatalyst layer.  The information regarding the thickness of the layer and the distribution of CuOx in the thickness direction should be necessary.

2.                 In Figure 6, the reviewer could not agree with the conclusion that “the optical bandgap energy changed by mixing TiO2 and CuOx.”  What transition did the decreased bandgap energy correspond to?  To prove the “decreased bandgap energy,” action spectra should be necessary.

3.                 In Figure 7, the authors mentioned that “a negative value of the potential indicates the accumulation of electrons in the material, while a positive value indicates the accumulation of holes (line 237-238).”  This is incorrect, because the OCP indicates just a potential difference from the Ag/AgCl electrode.  Thus, it does not indicate the accumulation of electrons or holes, or whether the n-type or p-type semiconductivity.

4.                 Regarding the photocatalytic CO2 reduction experiments, the isotopic labelling measurements using 13CO2 and the control experiments without light irradiation should be necessary.

5.                 What is the oxidative product? Oxygen evolved from water?

6.                 The functions of TiO2 and CuOx were unclear.  How did the authors calculate the energy diagram presented in Figure 9?  The figure was illustrated as if the reaction proceeded via the two-step photoexcitation process.

7.                 The present specimen should be referred to as mixture, rather than “composite.”

8.                 The almost same sentences were repeated in line 181-185 and line 186-190.  Please carefully check the manuscript before the submission.

Reviewer 5 Report

In this manuscript, the TiO2-CuOx compoiste materials were prepared and optimized, and the target manterials show the increased photocatalytic activity. I recommend the acceptance of the manuscript after reflecting the following comments.

1. The abstract should be carefully revised.

2. TEM was applied to characterize the morphology, so the elemental mapping and EDX should be provided.

3. Some minor errors, like the subscript, the format of Figure or Fig should be consistent.

4. In the manuscript, 'the figure' in line 181 and 186 should be given the exact Figure number.

5. For ref, some old references should be replaced by the recently references from 2020 to 2023. 

The spell or grammar should be carefully checked.

Round 2

Reviewer 4 Report

Re-comment 2 and 7: To verify the functions of TiO2 and CuOx as well as their interaction, action spectra are necessary.

Re-comment 3: Open-circuit potential under dark condition (OCPdark) reflects the equilibrium between the electrolyte redox potential and Fermi level of the semiconductor. The reference cited by the authors (Chem. Mater. 2016, 28, 12, 4231) also mentioned that the OCPdark significantly changed by adding some sacrificial reagents in the electrolyte. Therefore, difference in OCPs with and without light illumination does not represent the difference in the quasi-Fermi levels of holes and electrons.

Re-comment 5: Verification of oxidation reaction is quite important from the viewpoint of mass-balance and charge-balance. This is especially because photocatalytic CO2 reduction using water as an electron source is actually challenging reaction.

Author Response

Re-comment 2 and 7: To verify the functions of TiO2 and CuOx as well as their interaction, action spectra are necessary.

Response: 
Thank you for your comment. Indeed, conducting an action spectrum study would offer more comprehensive insights into the mechanism of photocatalytic reduction of CO2 involving TiO2 and CuOx, which holds scientific significance. However, in our opinion, such a study constitutes a separate and extensive endeavor that necessitates ample time, specialized equipment, and surpasses the scope of the current work being presented. Simultaneously, from a practical perspective, our study primarily focused on the full solar spectrum as it is of significant interest.

Re-comment 3: Open-circuit potential under dark condition (OCPdark) reflects the equilibrium between the electrolyte redox potential and Fermi level of the semiconductor. The reference cited by the authors (Chem. Mater. 2016, 28, 12, 4231) also mentioned that the OCPdark significantly changed by adding some sacrificial reagents in the electrolyte. Therefore, difference in OCPs with and without light illumination does not represent the difference in the quasi-Fermi levels of holes and electrons.

Response: 
As the esteemed reviewer correctly noted, the presence of sacrificial reagents adsorbed on the photoelectrode surface and reacting with the photogenerated charge carriers affects the measured OCP potential in the dark and under illumination. The OCP change is related to a wide range of electrochemical and electrophysical phenomena occurring on the surface and in the semiconductor photoelectrode volume, including those associated with changes in the concentration and ratio of charge carriers under light irradiation. We did not add sacrificial reagents to the electrolyte. In our study, we focus only on the dynamics of OCP changes under illumination with increasing amounts of copper oxide. It is the dynamics of OCP changes, in our opinion, that indicate a change in the concentration ratio of the charge carriers. We repeated this experiment several times using different samples. Yes, there were differences in the OCPdark for various samples. However, the dynamics of OCP changes under illumination were the same.

Re-comment 5: Verification of oxidation reaction is quite important from the viewpoint of mass-balance and charge-balance. This is especially because photocatalytic CO2 reduction using water as an electron source is actually challenging reaction.

Response: 
The esteemed reviewer rightly noted that photocatalytic reduction of CO2 using water is a complex reaction. Undoubtedly, verification of oxidation reaction is quite important from the viewpoint of mass-balance and charge-balance. Actually, the complexity of the reaction and its dependence on various factors prevents us from quickly obtaining a complete picture of the photocatalytic reduction of CO2 using water, including the oxidation reaction. We have plans for special studies of the oxidation reaction, which requires a series of experiments, but this is beyond the purpose of this article.